# Effect of LED Lighting on Selected Quality Parameters of Electricity

**DOI:** 10.3390/s23031582

**Published:** 2023-02-01

**Authors:** Agnieszka Wantuch, Michał Olesiak

**Affiliations:** Department of Electrical Engineering and Power Engineering, AGH University of Science and Technology in Krakow, al. Mickiewicza 30, 30-058 Kraków, Poland

**Keywords:** harmonic, power quality, LED lights

## Abstract

Recently, various technological light sources have appeared on the market. As a result, it is extremely important to evaluate the characteristics of the available lamps and the impact they may have on the power grid. This article presents regulations and standards for power quality, including higher current and voltage harmonics, and analyzes three selected light-emitting diode lamps. The purpose of the study was to analyze the impact of LED lighting systems on the power quality (PQ) of the electric grid. The results of measurements of parameters determining the power quality of the electricity consumed by modern LED light sources used for room lighting are presented. Commercially available lamps and low-power power supplies (25 to 35 W) with an output current of 350 mA were used for the study. The current waveforms of selected lamps during their connection to the power grid, the results of harmonic emission tests, as well as the effect of increasing the load of selected DC power supplies (drivers) on THD, power factor (PF), and their efficiency are presented. The significant negative impact of power supply circuits in compact LED lamps on the power quality parameters of lighting circuits was demonstrated. The tested lamps significantly exceed the permissible limits of harmonics and THD. Test results have shown that LED lamps show significant savings in electricity consumption, but they behave as nonlinear loads. They generate high-frequency current harmonics, which can lead to degradation of power quality in the distribution network. Therefore, if the primary concern for the user is power quality rather than power savings, traditional incandescent lamps would be a much better choice. When using switched-mode power supplies, attention should be paid to their load rating. You should avoid using power supplies in the lower ranges of permissible load capacity. A more heavily loaded power supply has better performance due to power quality and efficiency.

## 1. Introduction

LED lamps have been largely replacing traditional incandescent bulbs in both households and industry, road lighting, etc. They are also successfully replacing discharge lamps [1,2]. Due to their low power consumption and ability to produce colored light, they are also often used as decorative lighting. Lighting accounts for 19% of the world’s total electricity consumption, so lighting energy efficiency is crucial. Switching to LED lighting in the healthcare sector, education, or road lighting can generate energy savings of up to 80% [3]. An EU regulation banning the sale of incandescent lamps above 7 watts also went into effect in 2012 [4].

The first LED lamps that could replace traditional incandescent lamps appeared on the market as early as the late 1990s, but their light did not yet provide adequate visual comfort. Satisfactory results were obtained only at the beginning of the current century. At present, most of the lamps produced achieve luminous efficacy from 70 even to 205 lm/W [5].

Although LED lamps are characterized by high efficiency and long life, they also have their disadvantages—they cause an increase in reactive capacitive energy consumption and draw distorted current, which causes disturbances in the form of higher harmonics of current and thus deterioration in the quality of electricity [6,7]. The concept of power quality is a set of electrical quantities, such as voltage, frequency, harmonics, etc., which determine whether the electricity supplied to consumers maintains the appropriate parameters that allow the correct operation of all electrical equipment. Maintaining the appropriate quality of electricity necessitates meeting certain minimum requirements, which are written in Standards and Directives.

Deterioration of the quality of electricity can cause malfunction of equipment used to distribute electricity, as well as large disruptions in the operation of equipment connected to the power grid, while causing an increase in energy consumption. Therefore, with the growing popularity of LED lamps, it is becoming necessary to work on reducing the negative impact of LEDs on the power grid.

This paper presents a study of three selected commercially available LED lamps (A60 Graphene LED lamp E27 8W, Osram Parathom LED E14 6W, V-TAC E27 7W) and three low-power LED power supplies (OPTOTRONIC OT FIT 35/220-240/350, TCI ATON PRO 30 350-725mA, TRIDONIC lc 25/200-350/70 flexcc Ip SNc3). The purpose of the study was to analyze the impact of LED lighting systems on the power quality (*PQ*) of the electrical grid. The charts illustrate higher current harmonics, current and voltage waveforms, and the effect of power supply load on *THD*, *cosϕ,* and efficiency.

## 2. Laws Stipulating the Quality of Electricity

Model power quality occurs when the voltage curve is sinusoidal with a rated frequency (in Europe, 50 Hz) and an rms value equal to the rated voltage. However, this condition is impossible to achieve in reality.

The task of the energy supplier is to maintain the frequency and voltage values at the appropriate level. Power quality is influenced by both suppliers and consumers. Certain quantities describing power quality are subject to standardization processes [8]: mains frequency, voltage deviation, voltage fluctuation, flicker index, voltage asymmetry, voltage harmonics.

The basic legal act in the European Union is the EMC Directive 2014/30/EU [9], while in Poland the quality of electricity is defined by the Regulation of the Minister of Economy of 4.05.2007 on detailed conditions for the operation of the power system [10], which details the quality parameters of electricity that must be met by networks supplying consumers. These parameters include the following:the average value of the frequency and its maximum deviations,RMS value of the supply voltage and its maximum deviations,maximum indicator of long-term flicker *P*_lt_,average RMS value of the symmetrical component of the opposite order of the supply voltage to the value of the compatible order component and its maximum deviations,permissible average RMS values for each harmonic,maximum distortion coefficient by higher harmonics of the supply voltage *THD_U_* (*THD_I_* coefficient is not specified),consumption by the consumer of active power not greater than the contracted power, with a maximum coefficient *tgϕ*.

Other selected regulations can be found in [11,12,13], among others. The standard [12] divides consumer appliances into four classes, where lighting appliances are included in Class C, but LED lamps are not specified.

Power factor (*PF*) values, which are defined [12] as follows (1):(1)$$PF=\frac{P}{S}=\frac{U_{1}\cdot I_{1}\cdot cos\varphi_{1}+\sum_{n\neq 1}U_{n}\cdot I_{n}\cdot cos\varphi_{n}}{\sqrt{{\left(U_{1}\cdot I_{1}\right)}^{2}+\sum_{n\neq 1}{\left(U_{n}\cdot I_{n}\right)}^{2}}}$$
where: *U*_1_—RMS value of the fundamental harmonic voltage, *I*_1_—RMS value of the fundamental harmonic current, *ϕ_n_*—the phase shift angles of the current and voltage waveforms of the individual harmonics, *n*—the harmonic number.

For luminaires, this factor is specified in EU regulations [14]:P≤2 Wno requirements2 W<P≤5 Wcosφ(PF)>0.45 W<P≤25 Wcosφ(PF)>0.5P>25 Wcosφ(PF)>0.9

Selected power quality parameters basis of [9] concerning the quality of energy supplied by power producers are as follows:the average value of the frequency measured for 10 s should be within the following range:
(a)50 Hz ± 1% (from 49,5 Hz to 50,5 Hz) for 99.5% of the week;(b)50 Hz + 4%/–6% (from 47 Hz to 52 Hz) for 100% of the week;the permissible deviation of the RMS value of the voltage is ±10%;the distortion factor by higher harmonic of the supply voltage *THD_U_*, taking into account higher harmonics up to the order of 40, should be less than or equal to 8%.

## 3. Higher Harmonics

Common appliances are mostly supplied with a sinusoidal voltage with a frequency of 50 Hz and an RMS value of 230 V. With an ideal signal with the given parameters, devices would achieve greater efficiency and reliability. However, it is very difficult—and often impossible—to maintain such a signal for a sufficiently long time. Therefore, a harmonic signal appears; that is, no longer non-sinusoidal, but still periodic [8]. The reason for the formation of higher voltage harmonics—the frequency of which is a multiple of the basic signal, i.e., 50 Hz—is the effect of receivers with non-linear current-voltage characteristics on the power grid.

In electrical engineering, current and voltage harmonics can be distinguished. The distortion of the waveform of the above quantities can be described in the time domain (adopted in the standard [15]) or frequency. The most complete information is provided by a set of numbers, which determines the orders, amplitudes, and phases of the individual harmonics [8].

One of the normalized values is the maximum distortion coefficient by higher harmonic of the supply voltage *THD_U_*, defined by the following formula (2):(2)$$THD_{U}=\sqrt{\overset{40}{\underset{n=2}{\sum }}{\left(U_{n}\right)}^{2}}$$
where: *U_n_*—RMS value of the *n*th harmonic voltage, *n*—harmonic number.

The distortion factor of harmonic currents can be determined from the following formula (3):(3)$$THD_{I}=\frac{\sqrt{\sum_{n=2}^{40}I_{n}^{2}}}{I_{1}}$$
where: *I_n_*—RMS value of the *n*th harmonic of the current, *I*_1_—RMS value of the fundamental harmonic of the current, *n*—the harmonic number.

The permissible values of higher harmonics generated into the mains by electrical and electronic equipment with a phase supply current of up to 16 A are specified in the standard [12]. The permissible level of distortion of the current and voltage waveforms depends on the sensitivity of the supplied consumers.

## 4. LED Lighting

An LED is a semiconductor that emits light when the appropriate voltage is applied.

Despite the fact that LED lighting has many advantages—small size (so it can be used in many places inaccessible to traditional lighting), high efficiency, focused beam of light, resistance to damage, long life, no toxic components, low operating temperature, easy control, or the ability to shine in the color of your choice—at the same time, it is also not without disadvantages. The use of LED lighting is associated with the generation of higher current harmonics by these lamps, the consumption of capacitive reactive energy from the grid, the deterioration of the power factor, etc. [16,17].

A typical LED lamp design is shown in [18]. LEDs require the so-called drivers, i.e., control and power circuits, so dedicated integrated circuits are manufactured to power LED lamps. The block diagram of the LED lamp with an integrated driver is shown in Figure 1.

Two power supply methods are used: voltage and current. In a voltage supply, the voltage is stabilized, while the current depends on the resistance of the load. In current power supply, the current is stabilized by appropriate changes in voltage. It is very important to choose the right power supply, on which the proper operation of the lighting system and its durability can largely depend.

## 5. Measuring Apparatus and Test Method

Electroluminescent light sources can generate higher-harmonics currents, even far exceeding the normatively permissible levels. To check the quality of selected LED lamps on the market, appropriate measurements were made. Tests of electrical parameters determining the quality of electricity were carried out on a complete receiver, i.e., LED module with power supply.

The test stand allows testing of power quality parameters when powering selected light sources. The stand (Figure 2 and Figure 3) and tests were performed in accordance with the PN-EN IEC 61000-3-2: 2019-04. The tests were carried out under standardized atmospheric conditions, i.e., the air temperature was in the range of 20–27 °C. Measurements were performed on new sources. The lamps were lit until stabilization, which was taken as no change in the consumed power by more than 1% in 5 min.

To test selected low-power lighting sources in an isolated power supply system, the measurement system shown in Figure 3 was used. The measurement station consisted of the following:computer with dedicated software;Power and Harmonics Analyzer with Flickmaster HA1600A [19], along with dedicated software;AC1000A LOW DISTORTION POWER SOURCE [19];element under test.

The HA1600A is a fast and accurate AC current analyzer for single-phase loads up to 16 amps RMS. Dual power cables allow the load to be powered independently from the instrument’s power supply. Output to the load is via a “standard” power connector mounted on the front panel. A wide range of power connectors is available, including most domestic types. The HA1600A can measure watts, VA, rms volts, peak volts, rms amps, peak amps, peak factors, *THD*, power factor, frequency, and inrush current. The HA1600A includes RS232, USB interfaces for computer use, and a Centronics parallel interface for direct connection to a printer. The instrument’s firmware is stored in flash memory and can be updated via RS232 or USB as standards requirements evolve. The supplied HA-PC Link Plus software helps users perform routine compliance measurements and archive results. Data can be as a single report or can be continuous, allowing real-time viewing on a computer. Harmonics can be displayed on a computer as tabular reports or graphical histograms [19].

According to the standard [20], there are three criteria relating to luminaires whose power is in the range 5 ≤ P [W] ≤ 25. The tested luminaires were assigned to Class C. For a lamp to be approved for use, it must meet one of the three criteria tested.

In order to reduce errors that could result from temporary instability of one of the components of the test system, measurements for each LED lamp were repeated several times. This reduced the randomness of the test results, which could have occurred with only one series of measurements.

## 6. Studies of Selected Light Sources

The subject of the study is the control and power systems of compact LED lamps, which are replacements for incandescent light sources, of similar power, made on E27 and E14 caps. Each lamp is made by a different manufacturer. In this type of lamp, driver circuits are integrated with the LED module in a single fixture. The influence on the degree of distortion of the current waveform of a compact LED lamp is mainly due to the driver circuit, while the influence of the LED modules themselves is negligible.

Commercially available lamps were selected for the study: the A60 Graphene LED lamp E27 8W, Osram Parathom LED E14 6W, V-TAC E27 7W. The Osram and V-TAC lamps have similar luminous parameters and price, and the SERA’s lamp has a much higher luminous flux value and is almost twice as expensive.

Table 1 gives the basic technical data declared by manufacturers on the packaging or housings of the tested lighting sources.

All measurements were conducted at a base component level (50 Hz) of 230 V.

### 6.1. A60 Graphene LED Lamp E27 8W Lamp

Graphene LED Technology lamp datasheet available in [21]. 

Figure 4 shows the results of measurement for harmonics from 3 to 39. The black line indicates the permissible value, orange color - measured values.

The largest value is reached by the 3rd harmonic. The subsequent odd harmonics have smaller values. The limits for harmonics 9 and 23 were exceeded, but the current value of these harmonics is less than 5 mA, so the overall result of the test was considered positive. The value of the harmonic content index *THD_I_* is at 55.8%. For a lamp declared at 8 W, 7.6 W was measured.

The voltage (red color) and current (blue color) waveforms of the tested lamp are shown in Figure 5.

You can see that the current waveform (blue line) is not sinusoidal.

### 6.2. Osram Parathom LED E14 6W Lamp

The parameters of the lamp are available in [22].

Figure 6 shows the measurement results for harmonics from 3 to 39. The limits for harmonics 11 to 37 are exceeded. The *THD* coefficient is 30.2%.

Figure 7 shows the voltage and current waveforms of the tested lamp.

The waveform of the current drawn by the light source is far from a sinusoidal waveform, you can see spikes in the current drawn especially at the point where it passes 0, but the luminaire passes all tests with a positive result. In the test for harmonic emissions, all exceeded limits are less than 5 mA.

### 6.3. V-TAC E27 7W Lamp

Technical data are presented in [23].

Figure 8 shows the harmonic current emission graph of the tested luminaire. All harmonic limits were exceeded. The *THD* factor is 78.4%. The black line indicates the permissible value, orange color - measured values.

Figure 9 shows a voltage and current diagram for the tested lamp.

Harmonics from 9 to 19 were exceeded more than three times. An analogous situation occurs during the *THD* test. However, the waveform test has a positive result. Observing the waveform, it can be observed that the current waveform reaches a value of 5% of the maximum before 60°, the peak is reached before 65°, and the current value decreases to 5% before 90°. Thus, the light fixture has been approved for sale.

Tests conducted on LED lamps show that most of the light sources comply with the standards, despite the fact that some of them significantly exceed the permissible limits of harmonics and *THD*.

### 6.4. Summary of Results

Tests and analysis of measurements, carried out for three randomly selected compact LED lamps with integrated power supply systems, showed that such receivers can negatively affect the quality of electricity in power supply circuits. The main reason for this is that these light sources generate higher current harmonics, often far exceeding the permissible values. Of the lamps tested, only one met the requirements of the standard [12] for permissible levels of harmonic emissions. One lamp exceeded the limits for all harmonics, yet was still allowed to be sold, due to a positive waveform test.

Table 2 shows a comparison of *THD_I_* and the numbers of exceeded harmonics in the tested lamps.

Measurements show that these lamps consume odd harmonics, and *THD_I_* does not exceed 79%, which is a good result, since in some lamps *THD_I_* can exceed even 140% [16]. The lamps draw capacitive reactive power.

## 7. Studies of Selected DC Power Supplies

The nature of the load and the shape of the current waveform are strongly influenced by control and power supply systems using pulsed DC-DC converters.

In this study, we used pulsed DC power supplies, which are commonly used to power LEDs. Switching power supplies are characterized by small size, high efficiency, and high reliability. During the tests, such coefficients as harmonic content index (*THD_I_*), power factor (*cosϕ*), peak factor (*k*_pk_), and efficiency (*η*) were checked. Graphs of the respective load dependencies were made for the above quantities.

Power supplies from the following three manufacturers were used for the study: Tridonic, TCI, and OSRAM. The power supplies were loaded with the same LED modules over the entire operating range—from the minimum output voltage allowed by the manufacturer, up to the allowed maximum voltage. These values are closely related to the power of the power supply under test. Summary statements were made for an output current of 350 mA. The results of testing the same power supply with different output currents were also compared.

All power supplies were tested on the same test bench to eliminate errors caused by differences in meter classes. Measurements were made after the power consumed had stabilized. To determine the efficiency of the power supply, the input voltage, input current, and output power were measured with the HA1600A instrument, which also measures other parameters.

The measurement station includes the following (Figure 10 and Figure 11):computer with dedicated software;Power and Harmonics Analyzer with Flickmaster HA1600A, along with dedicated software;AC1000A LOW DISTORTION POWER SOURCE;voltmeter;ammeter;load in the form of LED modules.

The efficiency of the power supply was determined by the ratio of the input power measured with the HA1600A meter to the output power calculated by the product of the current and output voltage (4):(4)$$\eta =\frac{P_{out}}{P_{in}}$$

### Results

The group tested were low-power power supplies (25 to 35 W) with an output current of 350 mA:OPTOTRONIC OT FIT 35/220-240/350 [24],TCI ATON PRO 30 350-725mA [25],TRIDONIC lc 25/200-350/70 flexcc Ip SNc3 [26].

Figure 12 shows the effect of power supply load on *THD*.

In each of the tested power supplies, a clear decrease in the generated higher harmonics was observed with increasing load. *THD* remained similar for each power supply with the same percentage of load.

Figure 13 shows the dependence of the power factor on the load.

OSRAM and TCI power supplies maintain similar power factor values, which increases as the load increases. The Tridonic power supply deviates significantly from the other two products tested.

Figure 14 shows the dependence of power supply efficiency on load.

The best efficiency of the three power supplies tested is that of Tridonic, despite having the worst power quality parameters. The TCI power supply from the measurement at about 85% of the maximum load decreased its efficiency, which is consistent with the data sheet available on the manufacturer’s website.

## 8. Discussion

In this study, attention was paid to non-linear low-power consumers, which include LED lamps. It was shown that the power supply systems of these compact light sources generate higher harmonic currents, the level of which far exceeds the permissible values, which in turn can adversely affect other electricity consumers.

LED lamps with integrated power supply circuits in many cases had high current distortion, because until a dozen years ago these circuits did not contain filters, which meant that the coefficient of higher harmonic content of current *THD_I_* for many such lamps could exceed as much as 100%. However, even today, you can find LED lamps for which *THD* reaches as high as 140%, and the phase shift angle between current and voltage for the fundamental harmonic is negative and is about −50° ÷ −60° [15]. In particular, low-cost power supplies integrated in LED lamps, which do not have advanced solutions for improving the power factor and suppressing the propagation of higher harmonics of current, cause the *PF* power factor to take even values below 0.5, and the *CFI* peak factor rises even above the value of 4.5, where for a sinusoid it should be 1.41. Such a system generates higher harmonics of current, the values of which far exceed the permissible levels, which are defined by the standard [12]. In the case of simultaneous use of multiple lamps with the above parameters, there can be a significant deterioration in the quality of electricity.

In high-quality LED lamps, the *CFI* factor reaches a value of the order of 1.7 (that is, close to the peak factor of a sinusoidal waveform), the *PF* power factor and *cosϕ* reach values above 0.9.

Most of the power supplies tested showed a significant effect of output current on *THD*, *cosϕ,* and efficiency. Increasing the output current causes the above parameters to improve. The more loaded a power supply is, the better its performance due to power quality and efficiency. Thus, when using switching power supplies, attention should be paid to the degree of load. You should avoid using power supplies in the lower ranges of permissible load capacity. It is better to use a power supply with a lower power rating, loading it closer to the maximum values quoted by the manufacturer, than to use a larger power supply and load it in the lower ranges of allowable capacity.

In view of the above conclusions, the increasing use of LED lamps for lighting is becoming a challenge for power services, since until now there has primarily been a need for reactive inductive power compensation in the electric power industry. The generation of higher harmonic currents into the grid, which—with the presence of resistive-inductive and resistive-capacitive loads (LED lamps)—can lead to the formation of current resonance, which can cause higher harmonic voltages to rise above the values allowed by standards, as well as damage to capacitive loads [16].

## 9. Conclusions

The above facts have been confirmed by research conducted, among others, by in the electrical engineering laboratory of the Department of Electrical Engineering and Power Engineering at the Faculty of EAIiIB AGH. The article presents the research results of only a few of the dozen or so tested lamps—both traditional light bulbs and LED light sources. Measurements were made of the current, voltage, active and reactive power as well as the *cosϕ* factor, the *THD_I_* current harmonic content factor. In addition, measurements were also made that do not fall within the scope of the tests related to the quality of electricity, however, they also provide the opportunity to compare selected light sources with each other due to visual values, such as lighting intensity, and to check the reliability of the parameters presented by the manufacturer on the lamp packaging (not on the full details are given on all packages) or in catalog notes.

## Figures and Tables

**Figure 1 sensors-23-01582-f001:**
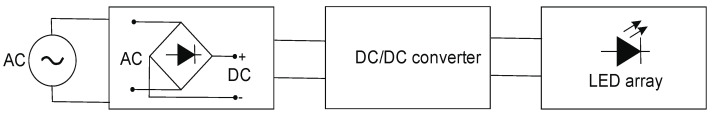
Block diagram of the LED lamp.

**Figure 2 sensors-23-01582-f002:**
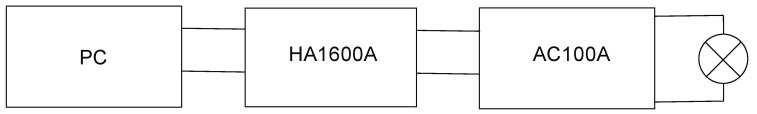
Schematic of the measuring station.

**Figure 3 sensors-23-01582-f003:**
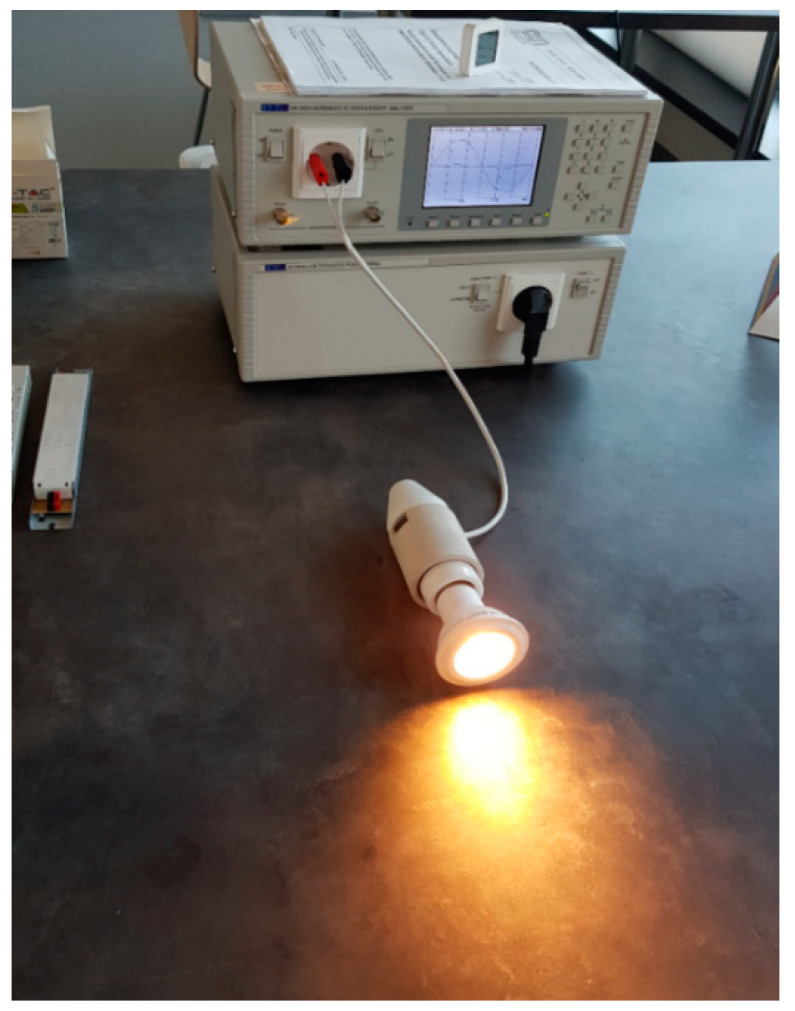
Photo of the measuring station with the tested source V-TAV PRO 7W E-27.

**Figure 4 sensors-23-01582-f004:**
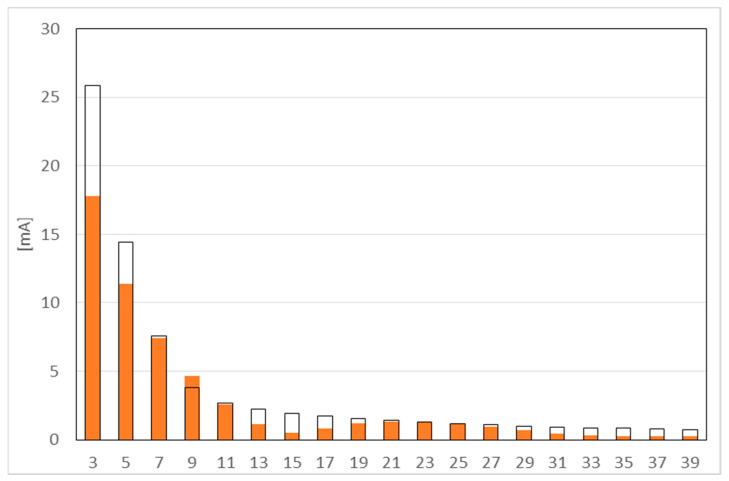
Harmonic emissions test.

**Figure 5 sensors-23-01582-f005:**
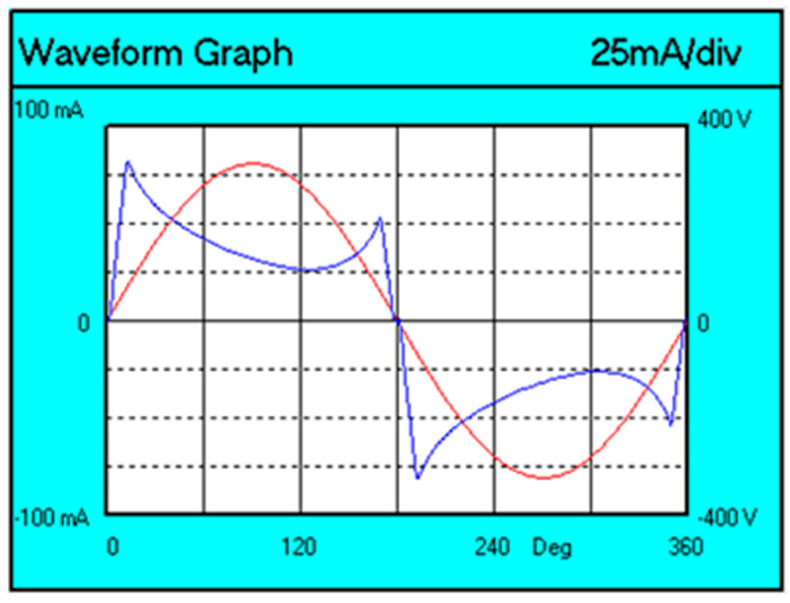
Voltage and current waveforms for the A60 Graphene lamp.

**Figure 6 sensors-23-01582-f006:**
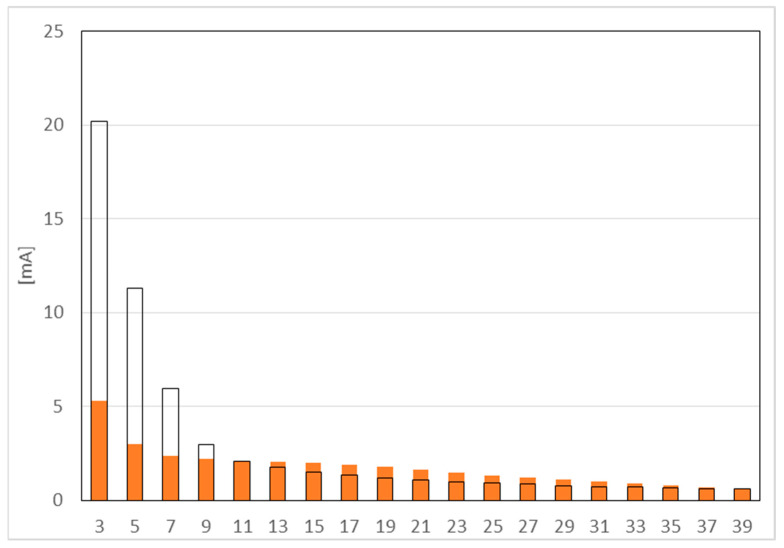
Harmonic emissions test.

**Figure 7 sensors-23-01582-f007:**
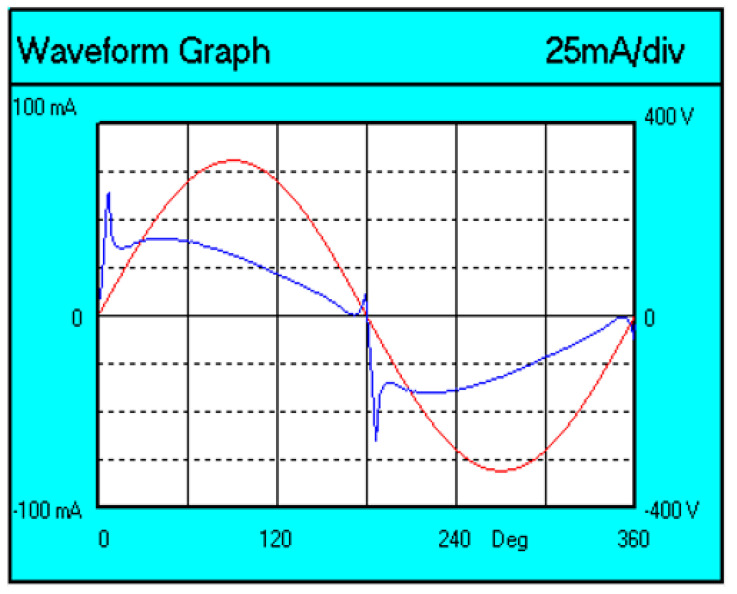
Voltage and current waveforms for the Osram Parathom lamp.

**Figure 8 sensors-23-01582-f008:**
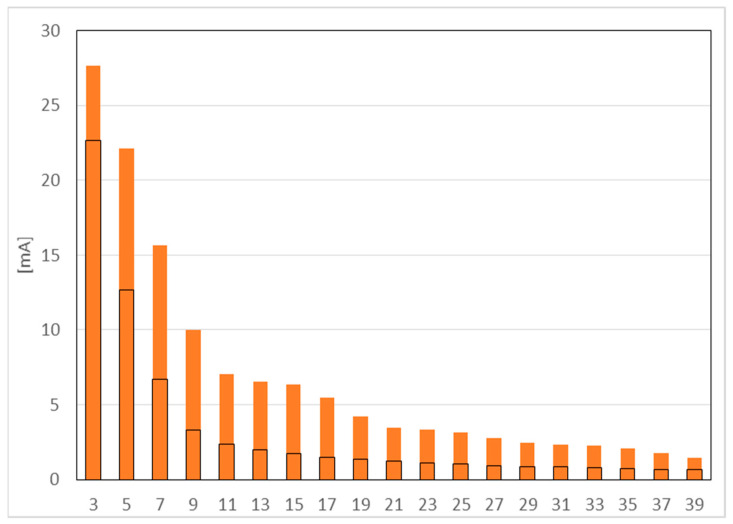
Harmonic emissions test.

**Figure 9 sensors-23-01582-f009:**
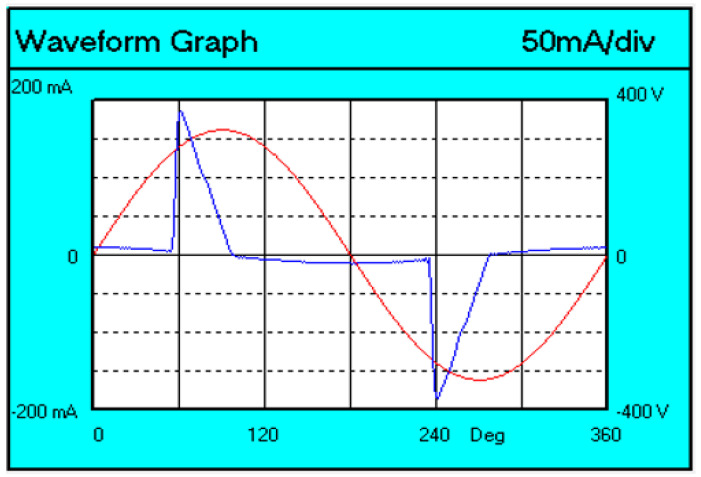
Waveform of voltage and current for a V-TAC lamp.

**Figure 10 sensors-23-01582-f010:**
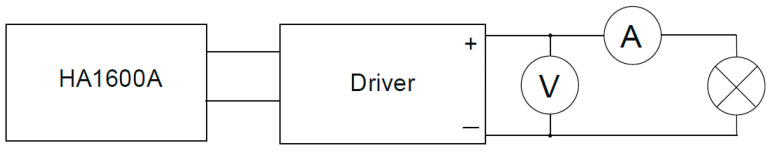
Scheme of the measuring stand.

**Figure 11 sensors-23-01582-f011:**
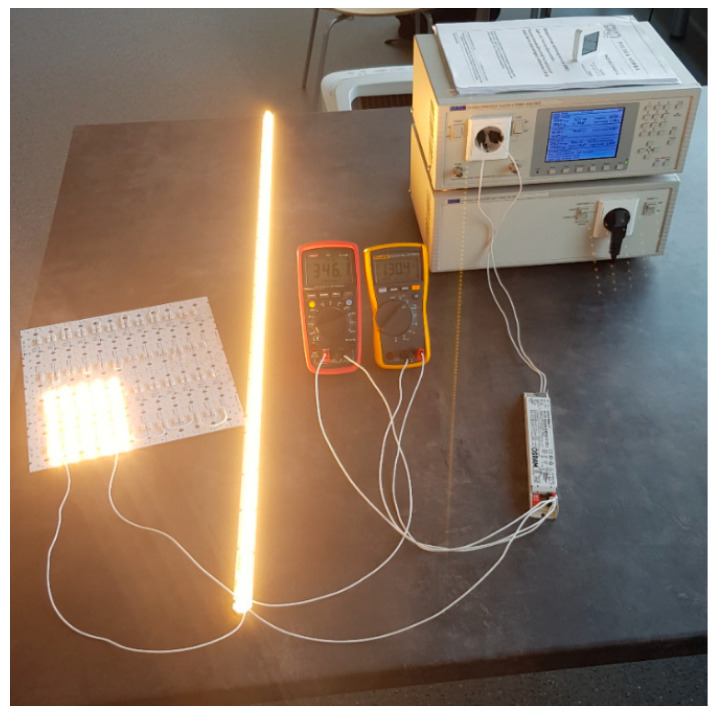
Photo of the station during measurements of LED power supplies. The photo shows the OPTOTRONIC OT FIT 65/220-240/350 D CS L power supply with load modules.

**Figure 12 sensors-23-01582-f012:**
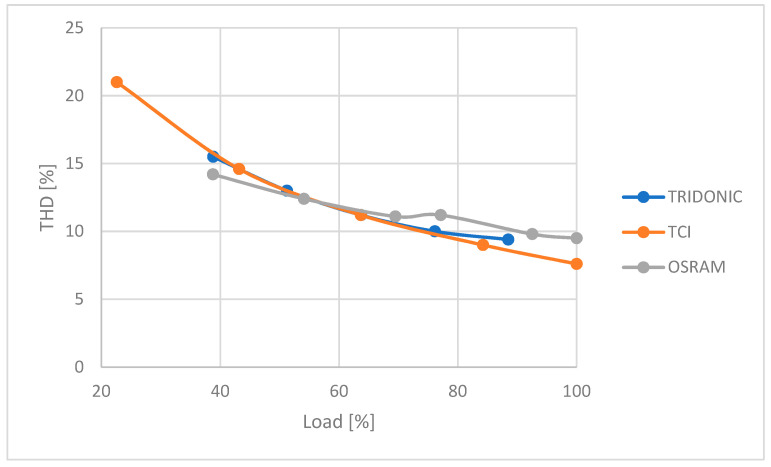
Influence of PSU load on *THD*.

**Figure 13 sensors-23-01582-f013:**
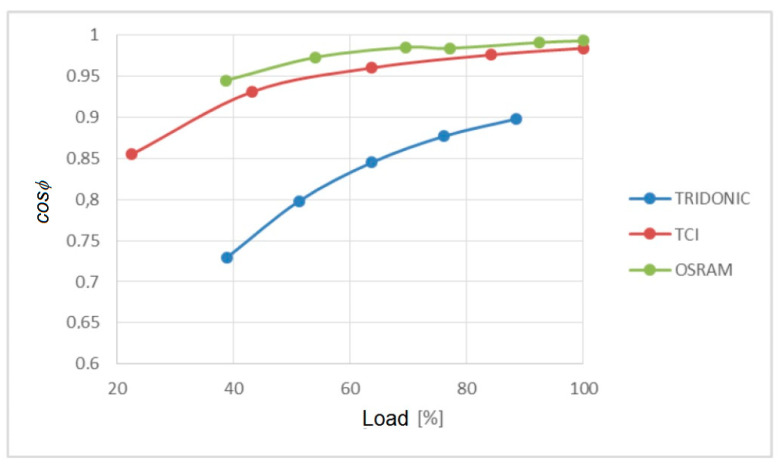
Impact of load increase on *cosϕ*.

**Figure 14 sensors-23-01582-f014:**
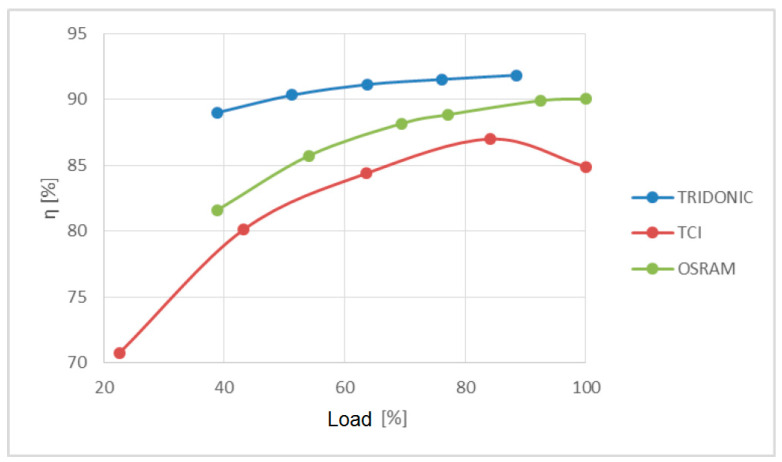
The influence of the power supply load on the efficiency.

**Table 1 sensors-23-01582-t001:** Declared parameters of low-power light sources.

Producer	Power [W]	Luminous Flux [lm]	Voltage [V]	PF	CRI	Thread
SERA	8	800	220–240	ND ^1^	>80	E27
OSRAM	6	470	220–240	0.5	>80	E14
V-TAC	7	495	220–240	>0.5	>80	E27

^1^ ND—value not declared by the manufacturer.

**Table 2 sensors-23-01582-t002:** THD and harmonic numbers that were exceeded.

Producer	Power [W]	THD [%]	Harmonic Numbers
SERA	8	55.8	9, 23
OSRAM	6	30.2	11–37
V-TAC	7	78.4	1–39
Kanlux ^1^	4	139	
PHILIPS ^2^	4	142	
LETHE ^3^	9.5	130	

Data from [14] ^1^—Kanlux DIXI COG4W: 4W; E27. ^2^—PHILIPS FILAMENT LED: 4W: 4W; E27. ^3^—LETHE LMP-G60: 9.5W; E27.

## Data Availability

Not applicable.

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
