# Peer review of "Effect of LED Lighting on Selected Quality Parameters of Electricity"

_sensors, 2023, doi:10.3390/s23031582_

Round 1

Reviewer 1 Report

The Abstract should be enlarged. Please indicate a purpose and describe the results obtained.   Please describe clearly the goals and objectives at the end of the Introduction chapter.   The review is very brief and does not fully cover the topic.   The article does not contain the section "Materials and Methods", it is necessary to add the chapter and describe there in detail technology and equipment used.   How many measurements were taken and in what repetition?   Why were these lamps and power supplies chosen for research?   The proposed data is the average of several measurements or it’s only one?   The figures should be presented in a higher quality, the symbols are hard to read.   The Discussion chapter is very brief, the results obtained should be compared with other studies.   Please clearly formulate the conclusions at the end of the study.  Specify for whom the results of the study will be useful.

Reviewer 2 Report

The manuscript entitled “Effect of LED Lighting on Selected Quality Parameters of Electricity” required the following major revisions

·         Author should be mentioned all obtained results in the abstract.

·         Author should be mentioned the scientific significance of the proposed work as qualitative results.

·         Comparison table of obtained results with other published results

·         Improve all the figures

Sentence formatting and English must improve.

·         Add more recent articles. 

Round 2

Reviewer 1 Report

The authors did not correct all the comments.

The review is very brief and does not fully cover the topic.  The introduction does not provide sufficient background and does not include all relevant references. The introduction should briefly place the study in a broad context and highlight why it is important.

Reviewer 2 Report

All the comments are satisfactory justify in the revised manuscript. Now the revised manuscript is recommended for acceptance.

Author Response

Thank you for the recommendation.